

# A first trial to supplement local hardwood sawdust at the first soaking for the cultivation of Shiitake (*Lentinula edodes*)

Stephanie Nabhan[1,2], Soukayna Haidar Ahmad[1], Zeina El Sebaaly[1], Teodor Nedelin[3] and Youssef Najib Sassine[1]

[1] Department of Plant Production, Faculty of Agriculture, Lebanese University, Beirut, Lebanon
[2] Department of Agronomy, Faculty of Agronomy, The University of Forestry, Sofia, Bulgaria
[3] Department of Forestry, Faculty of Forestry, The University of Forestry, Sofia, Bulgaria

Corresponding author
Zeina El Sebaaly,
zeina.sebaaly@st.ul.edu.lb

## ABSTRACT

Supplementation of the growing substrate has been reported to enhance the production of cultivated mushrooms; however, supplementation using nano-sized additives is not yet investigated on Shiitake (*Lentinula edodes*). The study investigated the potential of a nano-supplement (Lithovit®-Amino25) containing an admixture of 25% L-amino acids on shiitake cultivated on sawdust from locally available oak, maple, and apple trees (oak sawdust: OS, maple sawdust: MS, and apple sawdust: AS). Initially, sawdusts were enriched equally with wheat bran (WB) and at the first soaking, Lithovit®-Amino25 was applied at 5 g/L. Treatments were: T1: OS-WB (control), T2: OS-WB+ nano-amino, T3: MS-WB, T4: MS-WB+ nano-amino, T5: AS-WB, and T6: AS-WB+ nano-amino. Among non-supplemented substrates, complete mycelia run, fruiting, and harvest dates were faster in T1 than in T3 and T5. Complete mycelial development was delayed by 7, 5, 9, and 6 d in T3, T4, T5, and T6 compared to T1. The harvest date was delayed by 7.7–8.3 d on maple sawdust and by 10.5–12.7 d on apple sawdust compared to oak sawdust. However, nano-supplementation hastened fruiting and harvest dates (by 9.3 d) in T4 compared to T3. The biological yield of the second harvest was higher on supplemented than on non-supplemented maple and apple sawdusts. Only T1 and T2 showed consistency in production over two consecutive harvests. Nano-supplementation improved the total biological yield in T2, T4, and T6 by 9.8, 21.0, and 22.5%, respectively. Nevertheless, all treatments, except T4, had lower biological efficiencies compared with T1. In T4, results of stepwise regression showed a strong positive correlation ($R^2$ = 0.96) between the total biological yield and mushroom weight at the second harvest. Supplementation caused a slight or significant improvement in pileus diameter and mushroom firmness and a significant improvement in mushroom's crude protein (by 2.9–8.2% compared to T1) and fiber contents (by 1–2.3% compared to T1). In conclusion, supplemented maple sawdust would alternate oak sawdust for shiitake production, though other timings of supplementation might be further investigated to optimize production on this substrate.

# INTRODUCTION

*Lentinula edodes* (Berk.) Pegler, commonly known as shiitake mushroom, belongs to the species edodes and the genus *Lentinula* (*Annepu et al., 2019*). According to *Zhao et al. (2019)*, shiitake represents 25% of the world's mushroom production and is the second most produced edible mushroom. It is characterized by its unique, intense, and meat like flavor, described as 'umami' taste (*Shi et al., 2020*), and its nutritional composition being rich in vitamins (B1, B2, B12, E, C, and D), minerals (Ca, K, Mg, P, Na, Cu, Fe, Mn, Zn, Se), macronutrients, sugars, tocopherols, and polyunsaturated fatty acids (*Reis et al., 2012*; *Heleno et al., 2012*). Also, shiitake is known for its anti-cancer, antioxidant, anti-inflammatory, and hypocholesterolemic properties, being rich in polysaccharide (ß-glucan) and bioactive compounds (lenthionine, eritadenine, and lentinan) (*Petravić-tominac et al., 2010*; *Yukawa et al., 2012*; *Finimundy et al., 2014*; *Valverde, Hernández-Pérez & Paredes-López, 2015*; *Wang et al., 2023*). In the wild, shiitake is a wood-decaying fungus that grows on dead logs of various broadleaf trees, like *Quercus* spp., *Castenopsis* spp., *Elseocarpus* spp., *Lithocarpus* spp., *Betula* spp., and *Carpinus* spp. (*Eira, Meirelles & Paccola Meirelles, 2010*). Synthetic logs of oak sawdust are generally used for commercial cultivation of the mushroom. Sawdust of maple, birch, poplar, or other similar hardwood trees (*Menolli et al., 2022*) can be used as well (*Gaitán-Hernández et al., 2020*). Practically, the standard substrate formula applied at industrial scale is composed of 80% hardwood sawdust with 20% starch-based supplements (wheat bran, rice bran, millet, rye, maize powder, sunflower hull, soybean flour, *etc.*) (*Ramkumar, Thirunavukkarasu & Ramanathan, 2010*). The choice of an appropriate substrate for a particular species of mushroom depends on the nutritional requirements of the species as well as availability of substrate materials in the intended cultivation area (*El Sebaaly et al., 2020a*, *2020b*; *Abou Fayssal et al., 2021b*; *Alsanad et al., 2021*). Early studies reported a variety of lignocellulosic materials that could potentially be used in shiitake cultivation as an alternative to oak sawdust, originating from agricultural activities or forests: sunflower seed husks (*Curvetto, Figlas & Delmastro, 2002*), hazelnut husks (*Özçelik & Pekşen, 2007*), cereal residues (*Philippoussis, Diamantopoulou & Israilides, 2007*), sugar maple (Acer sp.) (*Bruhn, Mihail & Pickens, 2009*), corncobs (*Yu et al., 2021*), red maple (*Acer rubrum*) (*Frey et al., 2020*), and apple sawdust (*Yu et al., 2022*).

The quantity, quality, and nutritional composition of mushrooms are significantly impacted by supplementation of the mushroom-growing substrate. To achieve the intended results, it is crucial to appropriately select the supplement type, time, and application method (*Carrasco et al., 2018*; *Naim et al., 2020a*; *Sassine et al., 2021*).

Supplements are applied during substrate preparation to promote the vegetative growth throughout the substrate (*Naraian et al., 2009*), or at the end of spawn run to enhance mushroom fructification (*Pardo-Giménez et al., 2016*).

Most of the early trials on shiitake employed starch- or protein-rich supplements added during substrate preparation and showing a positive effect on shiitake growth and yield, such as cotton seed meal, soybean meal (*Kapoor, Pardeep & Katyal, 2009*), wheat bran, rice bran, and maize powder (*Moonmoon et al., 2011*; *Ranjabar & Olfati, 2016*), corncobs (*Eira, Meirelles & Paccola Meirelles, 2010*). Also, vineyard pruning and olive pruning showed important effects on shiitake in the latest study of *Sassine et al. (2024)*. Nevertheless, rare studies investigated the supplementation of shiitake growing substrate at the end of the spawn run or the use of nano-supplements for this purpose. On the other hand, the safe use of a nano-supplement (Lithovit®-Amino 25) to produce *Pleurotus ostreatus* was reported in early studies by *Naim et al. (2020b)* and *Sassine, Shuleva & El Sebaaly, 2022*; the product applied at spawning and after the first harvest at a low dosage (3 g/kg) caused higher mushroom weight and yield, and improved nutritional composition.

The study evaluated the potential of readily available hardwood types (oak, maple, and apple) sourced from local forests or agricultural fields. It also marks the first trial to supplement the shiitake growing substrate using a supplement of nano-metric size (Lithovit®-Amino 25) applied at soaking and to investigate its effect on the growth, production, quality, and nutritional value of the mushroom.

# MATERIALS AND METHODS

## Experimental design and treatment

The experiment was arranged in a completely randomized design with six treatments and six replicates per treatment (six bags of one kg per treatment). Initially, each replicate contained 800 g of sawdust and 200 g of wheat bran. Sawdust was supplemented or not with nano-supplement and treatments were as follows: T1: OS-WB, T2: OS-WB+ nano-amino, T3: MS-WB, T4: MS-WB+ nano-amino, T5: AS-WB, and T6: AS-WB+ nano-amino.

## Evaluation of sawdust properties

Various analytical tests were performed to assess the physico-chemical properties of each sawdust type at the Lebanese Agricultural Research Institute (LARI)-Tal Amara station using a sample of 0.5 kg. Sawdust pH was evaluated on filtrates of sawdust samples using a pH-meter (ADWA AD 132). Crude protein was assessed using the Micro-Kjeldahl method (N × 6.25) (*Association of Official Chemistry (AOAC), 1995*). Moisture content determination followed (*Association of Official Chemistry (AOAC), 2005a*): two grams of substrate samples were dried under pressure of 13332.2 Pa to a constant weight at 95–100 °C. Ash content was determined according to the *Association of Official Chemistry (AOAC) (2005a)*: two grams of sample were weighed into a porcelain crucible and preheated to 600 °C for 2 h in a temperature-controlled furnace. Then, the crucible was transferred to a desiccator, cooled, and weighed, and the percentage of ash was recorded as follows:

$$Ash(\%) = \frac{\text{Initial weight of sample } (W_1)(g) - \text{weight of sample after ash drying } (W_2)(g)}{\text{Initial weight of sample } (W_1)(g)}$$

Crude fat was assessed by acid hydrolysis method according to the *Association of Official Chemistry (AOAC) (2005b)*. Dry matter, organic matter, carbon, nitrogen, and carbohydrates contents of sawdust were determined by calculation as follows:

$$\text{Dry matter } (\%) = 100 - \text{moisture } (\%)$$

$$\text{Organic matter } (\%) = 100 - \text{ash } (\%)$$

$$\text{Carbon } (\%) = \frac{\text{organic matter } (\%)}{1.72}$$

$$\text{Nitrogen } (\%) = \text{crude protein } (\%) \times 6.25$$

$$\text{Carbohydrates } (\%) = 100 - \text{moisture } (\%) - \text{crude protein } (\%) - \text{crude fat } (\%) - \text{minerals } (\%)$$

Crude fibers content was assessed based on the Weende method according to the *Association of Official Chemistry (AOAC) (2005c)*. Fiber fractions (acid detergent fiber: ADF, neutral detergent fiber: NDF, and acid detergent lignin: ADL) were measured by the Fibertherm methodology according to *Wannasawang et al. (2023)*. Consequently, the contents of cellulose, hemicellulose, and lignin were calculated as follows:

$$\text{Lignin } (\%) = \text{ADF } (\%)$$

$$\text{Hemicellulose } (\%) = \text{NDF } (\%) - \text{ADF } (\%)$$

$$\text{Cellulose } (\%) = \text{ADF } (\%) - \text{ADL } (\%)$$

## Substrates preparation

A Pro Wood chipper was used to reduce apple wood logs sourced from a local organic apple orchard to sawdust size. Oak and maple sawdust sourced from 'Mushies', a local mushroom grower, was initially collected from local forests. Sawdust of the three types underwent a first pasteurization for 15 min in boiling water at 100 °C to remove any dirt, insects, or any fungal or bacterial contamination. Then, the sawdust was drained and sundried for several days. To prepare each substrate, 4.8 kg of sawdust were mixed with 1.2 kg of wheat bran using a concrete mixer (74,540) to get a homogenous formula. Thereafter, substrate formulas were wetted to a moisture level of 60–70%; when the squeezed substrate expands slowly, no dampness appears on the hand (*Green & Mueller, 1995*). After wetting, substrate formulas were filled into six autoclavable polypropylene filter patch bags of 1 kg capacity each. The bags were then sealed and placed in modified barrel steamers equipped with a closing pressurized cover, thermometer, pressure gauge, and a faucet, and were subjected to steam pasteurization for 6 h at 120 °C and $10^4$ Pa.

## Cooling and spawning

After autoclaving, bags were transferred to a clean and cool room set at 16 °C to reduce the substrate temperature below 25 °C for an appropriate spawning process and suitable mycelium growth (*Chang & Miles, 2004*). Spawning (at a 2% rate) was carried out in hygienic conditions at the top of substrates using grain spawn of shiitake strain 3782 sourced from Mushies (an online store). Then, the bags were labelled and moved to the incubation room.

## Incubation phase

Incubation occurred in complete darkness at a temperature of 20–22 °C and a relative humidity of 50–60%. Temperature and relative humidity inside the incubation room were monitored and kept constant using the humidity/temperature meter (Lutron HT-3007SD). During this phase, five consecutive stages of mycelial development were distinguished, starting with stage 1: mycelia run (thin layer of white hyphae completely covering the block), followed by stage 2: mycelia coat formation (hardening of the mycelia sheet covering the whole substrate surface), stage 3: bump formation (clumps of mycelia developing into popcorn shape), stage 4: browning/pigmentation (mycelia block developing a dark brown and dry outer protective layer), and stage 5: coat hardening (hardening of the dark coat) (*Chen, 2001*). The end of each stage was recorded in days after spawning (DAS).

## Soaking and supplementation

When blocks hardened (stage 5), polyethylene bags were removed, and the substrate blocks were soaked in plastic bags containing ice for 24 h, creating a thermal shock to induce fruiting (*Stamets, 1993*). With respect to treatments T2, T4, and T6, 5 grams of Lithovit®-Amino 25 were added to 1 L of cold distilled water (concentration 5 g/L) and added to the soaking bag containing ice. The product is a nitrogen-rich fertilizer suitable for use in organic farming (according to Regulation (EC) No. 834/2007-European Community). It is made by tribodynamic activation and micronization of dolomite, and it is composed of 50.0% calcium carbonate ($CaCO_3$), 28.0% calcium oxide (CaO), 9.0% silicon dioxide ($SiO_2$), 3.0% total nitrogen (N), 1.8% magnesium oxide (MgO), 0.5% iron (Fe), and 0.02% manganese (Mn). It is obtained by adding (25% admixture) highly energised 16 water-soluble vegetable L-amino acids to Lithovit particles (*Bilal, 2010*).

## Fruit induction and harvest

Twenty-four hours after soaking, the fruiting room was set at a temperature of 16 °C, a relative humidity >90%, and artificial light (2000 lux) to induce fruit formation. High relative humidity was maintained inside the fruiting room using humidifiers (SANI-JET AIR 2836/P0). Two to three days later, pinheads started to appear; thus, blocks were watered daily, three times a day, until mushrooms became fully developed. The date of fruiting was recorded on all treatments as soon as distinct pinheads started to appear on the blocks. Fruit development took around four to five days at the different treatments. Mushrooms were harvested using clean scissors as soon as the gills became visible and the

outer edge of the mushroom was slightly curled under (*Mudge et al., 2013*). Figure 1 below shows the spawn run and fruiting stages.

The date of harvest was then recorded in DAS. After the first harvest, the blocks were put again in dark conditions for around 2 weeks. Two weeks later, blocks were soaked for a second time in ice for all treatments to induce a second harvest. The Fig. 2 below shows the readiness of mushrooms for harvest.

Mushroom number at first (MN1) and second (MN2) harvests and mushroom weight (MW1) and second (MW2) harvests were recorded on the total number of mushrooms harvested per bag. Biological yield corresponded to the total weight of mushrooms harvested per bag at harvest 1 (BY1) and harvest 2 (BY2). The total biological yield (TBY) corresponded to the summation of BY1 and BY2. Biological efficiency (BE) was calculated for each treatment according to *Yang, Guo & Wan (2013)*, as follows:

$$BE(\%) = \frac{total\ biological\ yield\ (g/bag)}{initial\ dry\ weight\ of\ substrate\ (g)x\ 100}$$

Further, the following physical characteristics were assessed on mushrooms obtained at harvest 1 using a sliding caliper: pileus diameter (PD), pileus thickness (PT), stipe diameter (SD), and stipe length (SL), and the ratio PD/SL was calculated. Mushroom firmness was measured using a setamatic penetrometer (Stanhope-Seta) on five different points of the pileus and the average value was calculated for each sample.

## Evaluation of mushroom composition

Various analytical tests were conducted on representative fresh samples of each treatment to evaluate their nutritional composition. A moisture analyzer (M5-Thermo A64M) was used to evaluate the moisture content. For the determination of ash content, five grams of macerated mushroom were heated in a muffle furnace (Carbolite Furnace OAF 10/1) at 550 °C for 24 h, and the ash content was calculated as follows:

$$Ash\ (g) = W_2(g) - W_1(g)$$

where W2: weight of the crucible containing ash, and W1: weight of an empty crucible.

Fat content was determined using the Soxhlet apparatus technique (*Luque de Castro & García Ayuso, 2000*) as follows: 10 g of mushrooms were macerated with sea sand and placed in the extractor. The round bottom flask of the extractor was weighed ($W_1$), then filled with 500 mL of hexane and heated. The cycle was repeated several times for 12 h, during which hexane gradually evaporated and the extracted fat remained at the flask bottom. The flask containing the extracted fat was weighed ($W_2$). Fat content was calculated and then expressed in percentage:

$$Fat\ (g) = W_2(g) - W_1\ (g)$$

Crude protein content (N × 4.38) (%) was determined using the macro-Kjeldhall method based on *Reis et al. (2012)*. Crude fiber content (%) was determined according to *Association of Official Chemistry (AOAC) (1982)* as the loss of ignition of dried residue
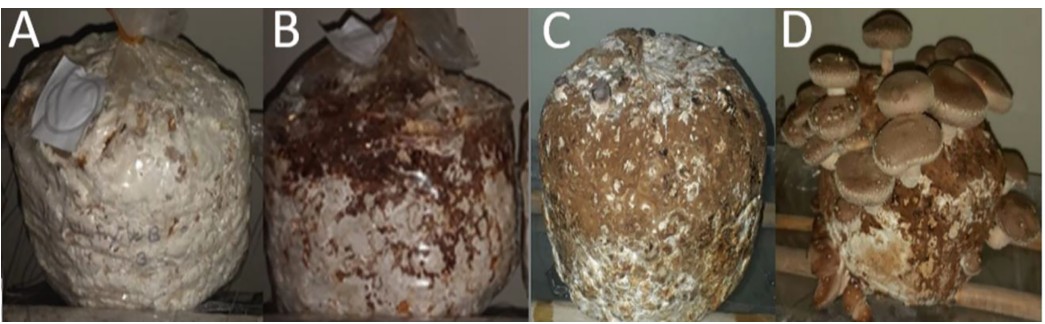

**Figure 1** (A) Bump formation, (B) coat hardening, (C) pinhead formation, (D) fruiting.

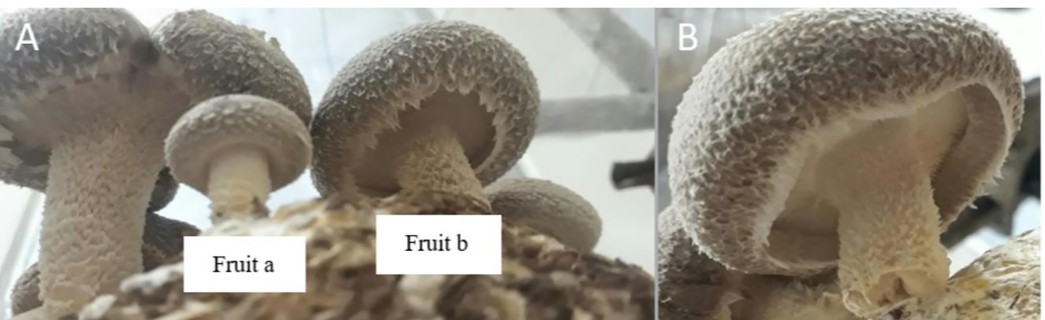

**Figure 2** (A) 'Fruit a' represents a mushroom that is not yet ready for harvest, 'Fruit 'b' represents a mushroom that is ready for harvest, (B) fruit ready for harvest.

remaining after digestion of the mushroom sample with 1.25% (w/v) $H_2HSO_4$ and 1.25% (w/v) NaOH. Carbohydrates content was then calculated as follows:

$$\text{Carbohydrates } (\%) = 100 - (\text{crude fiber } (\%) + \text{crude protein } (\%) + \text{fat } (\%) + \text{moisture } (\%))$$

Vitamin C as ascorbic acid (mg/100 g) was determined titrametrically using 2.6 Dichlorophenolindophenol methods based on *Tibuhwa (2014)*. A 5% metaphosphoric acid solution was added to a known weight of grounded sample, and the mixture was stirred for 30 min. The mixture was then filtered through Whatman No. 42 filter paper using a suction pump. 10 mL of the filtrate were pipetted into a 250 mL conical flask and titrated with 0.025% of 2.6 Dichlorophenolindophenol reagents. The amount of vitamin C in each extract was calculated using the following equation:

$$mg \; of \; ascorbic \; acid \; per \; 100 \; g = \frac{A x I x V_1 x 100}{V_2 x W}$$

where A = quantity of ascorbic acid (mg) reacting with 1 ml of 2.6 indophenol;

I = volume of indophenol (in mL) required for the completion for the titration with extract;

$V_1$ = total volume of extract;

$V_2$ = volume of extract used for each titration.

W = weight of the mushroom sample extracted

Vitamin $D_2$ as ergocalciferol and vitamin $D_3$ as cholecalciferol (in µg/100 g) were determined by high performance liquid chromatography (HPLC) according to *Mattila et al. (1994)*. Analytical tests on mushrooms were performed in triplicates.

### Statistical analysis

Data analysis applied the two-way ANOVA (factorial ANOVA) and means were compared by Duncan's Multiple Range test at $P_{value} < 0.05$ using SPSS 25 program version 26. Also, an independent sample t-test compared production between harvests 1 and 2. Stepwise regression evaluated the relationship between the total biological yield (dependent variable) and the productive indicators (MN1 and MN2, MW1 and MW2, BY1 and BY2, and PD, PT, SD, and SL) as independent variables.

## RESULTS

### Sawdust properties

Analysis of substrate properties (Table 1) showed that MS had a higher C/N ratio, thus a lower nitrogen content compared to OS and AS. Crude protein content was the highest in OS, followed by AS and MS (3.04, 2.84, and 1.59%, respectively). Also, MS was the richest in carbohydrates and the poorest in ash content, while OS was the richest in ash and the poorest in carbohydrates. While lignin content was lower in MS and AS than in OS (29.2, 21.6, and 17.7%, respectively), hemicellulose content was higher in both sawdust types than in OS (37.1, 16.4, and 9.1%, respectively). In addition, cellulose content was the highest in AS and lower in MS compared to OS (50.2, 29.2, and 38.8%, respectively). Sawdust pH was higher in MS and AS than in OS (6.6., 6.7, and 5.4, respectively).

### Shiitake growth on different substrates

Results (Table 2) showed that in comparison to the control (T1), there was a significant delay in the dates of the different stages of mycelia growth: stages 1, 2, 3, 4, and 5, in T3, T5, and T6, and in the stages 2, 4, and 5 recorded in T4. In particular, the last stage of growth (stage 5) showed respective delays of 6.6, 4.8, 9.3, and 5.6 d in T3, T4, T5, and T6 compared to T1. All stages of mycelia development were almost achieved simultaneously on maple and apple sawdust.

Findings in Fig. 3 showed a significant difference in the dates of fruit formation and harvest in supplemented maple and apple sawdust compared to non-supplemented cases, with an earliness in both dates in supplemented cases; by 4.1 and 3.6 d, respectively, in T4 compared to T3, and 5.4 and 5.2 d, respectively, in T6 compared to T5. Also, there was a slight difference in harvest date recorded on supplemented oak sawdust compared to non-supplemented substrate: earliness of 3.0 d. However, whether the three sawdust were supplemented or not, harvest on maple and apple sawdust was significantly delayed compared to oak sawdust. The delay in harvest ranged between 7.7 and 8.3 d on maple sawdust and between 10.5 and 12.7 d on apple sawdust compared to oak sawdust.

**Table 1 Properties of oak, maple, and apple sawdust.**

|  | OS | MS | AS |
|---|---|---|---|
| **Dry matter (%)** | 91.26 | 90.96 | 93.91 |
| **Moisture (%)** | 8.73 | 9.03 | 6.08 |
| **Carbon (%)** | 53.80 | 56.83 | 54.60 |
| **Nitrogen (%)** | 0.48 | 0.25 | 0.45 |
| **C/N ratio** | 109.8 | 227.3 | 121.3 |
| **Organic matter (%)** | 92.54 | 97.76 | 93.91 |
| **Crude protein (%)** | 3.04 | 1.59 | 2.84 |
| **Ash (%)** | 7.45 | 2.23 | 6.08 |
| **Fat (%)** | 0.74 | 0.84 | 1.86 |
| **Carbohydrates (%)** | 80.01 | 86.29 | 83.11 |
| **NDF (%)** | 77.18 | 87.84 | 84.23 |
| **ADF (%)** | 68.05 | 50.77 | 67.83 |
| **ADL = lignin (%)** | 29.2 | 21.6 | 17.7 |
| **Hemicellulose (%)** | 9.1 | 37.1 | 16.4 |
| **Cellulose (%)** | 38.8 | 29.2 | 50.2 |
| **pH** | 5.4 | 6.6 | 6.7 |

Note: OS, oak sawdust; MS, maple sawdust; AS, apple sawdust; NDF, neutral detergent fiber; ADF, acid detergent fiber; ADL, acid detergent lignin.

**Table 2 Effects of substrates and supplementation on the dates of mycelia developmental stages (in DAS: days after spawning) recorded on initial substrates formulas and on the productive indicators recorded at the first and second harvests.**

|  | T1: OS-WB | T2: OS-WB+ | T3: MS-WB | T4: MS-WB+ | T5: AS-WB | T6: AS-WB+ | P value |  |  |
|---|---|---|---|---|---|---|---|---|---|
| **Stage 1** | 15.7 ± 1.0bc | 14.2 ± 2.3c | 19.0 ± 2.6a | 18.0 ± 0.9ab | 18.5 ± 2.1a | 18.8 ± 3.2a | 0.000 |  |  |
| **Stage 2** | 22.8 ± 1.9b | 21.8 ± 2.7b | 27.8 ± 3.2a | 26.0 ± 0.9a | 26.3 ± 2.7a | 26.7 ± 3.3a | 0.000 |  |  |
| **Stage 3** | 37.3 ± 2.7bc | 36.0 ± 2.0c | 43.5 ± 5.0a | 41.0 ± 1.8ab | 42.5 ± 2.6a | 42.7 ± 3.5a | 0.000 |  |  |
| **Stage 4** | 61.3 ± 3.1b | 59.3 ± 1.9b | 69.5 ± 5.7a | 68.3 ± 2.8a | 70.5 ± 2.6a | 68.5 ± 2.4a | 0.000 |  |  |
| **Stage 5** | 72.2 ± 2.1c | 69.3 ± 1.6c | 78.8 ± 6.9ab | 77.0 ± 2.7b | 81.5 ± 2.1a | 79.8 ± 2.0ab | 0.000 |  |  |

|  | T1: OS-WB | T2: OS-WB+ | T3: MS-WB | T4: MS-WB+ | T5: AS-WB | T6: AS-WB+ | Sub | Sup | Sub x sup |
|---|---|---|---|---|---|---|---|---|---|
| **MN1** | 29.5 ± 3.5a | 11.3 ± 2.2c | 13.0 ± 2.1bc | 15.3 ± 1.9b | 10.3 ± 2.7c | 6.3 ± 2.6d | 0.000* | 0.000* | 0.000* |
| **MW1 (g)** | 8.8 ± 0.5d | 39.2 ± 1.6a | 21.6 ± 2.9c | 20.6 ± 2.1c | 20.9 ± 1.6c | 36.3 ± 2.6b | 0.000* | 0.000* | 0.000* |
| **BY1 (g/bag)** | 275.0 ± 52.2cd | 350.6 ± 28.9a | 291.8 ± 35.0bcd | 329.7 ± 10.3ab | 249.4 ± 55.9d | 312.8 ± 27.6abc | 0.093 | 0.000* | 0.478 |
| **MN2** | 24.0 ± 2.8a | 9.3 ± 2.3c | 9.5 ± 1.9c | 13.2 ± 2.8b | 4.8 ± 1.7d | 12.8 ± 4.8bc | 0.000* | 0.312 | 0.000* |
| **MW2 (g)** | 11.6 ± 3.3e | 28.1 ± 1.1a | 14.6 ± 0.5d | 17.5 ± 2.7c | 22.4 ± 2.6b | 20.2 ± 1.3b | 0.000* | 0.000* | 0.000* |
| **BY2 (g/bag)** | 322.0 ± 43.6a | 322.1 ± 39.8a | 130.6 ± 32.5d | 205.6 ± 30.7b | 103.6 ± 2.1d | 142.8 ± 5.3c | 0.000* | 0.001* | 0.019* |
| **TBY (g/bag)** | 597.0 ± 93.2b | 622.2 ± 47.6a | 422.4 ± 31.5c | 535.3 ± 23.6b | 353.1 ± 55.9d | 455.6 ± 28.4c | 0.000* | 0.000* | 0.510 |
| **BE (%)** | 59.7 ± 9.3b | 66.2 ± 4.8a | 42.2 ± 3.1c | 53.5 ± 2.4b | 35.3 ± 5.6d | 45.6 ± 2.8c | 0.000* | 0.000* | 0.510 |

Note:
Oak sawdust, MS, Maple sawdust; AS, Apple sawdust; WB, Wheat bran; +, nano-amino supplement added to the substrate; Stage 1, mycelial run; Stage 2, coat formation; Stage 3, bump formation; Stage 4, pigmentation; Stage 5, coat hardening; MN, mushroom number; MW, mushroom weight; BY, biological yield; BE, biological efficiency. Different letters in the same column indicate a statistically significant difference according to Duncan's multiple range test at P value <0.05. An asterisk (*) indicates a statistically significant effect of the studied factors on tested indicators.

2000

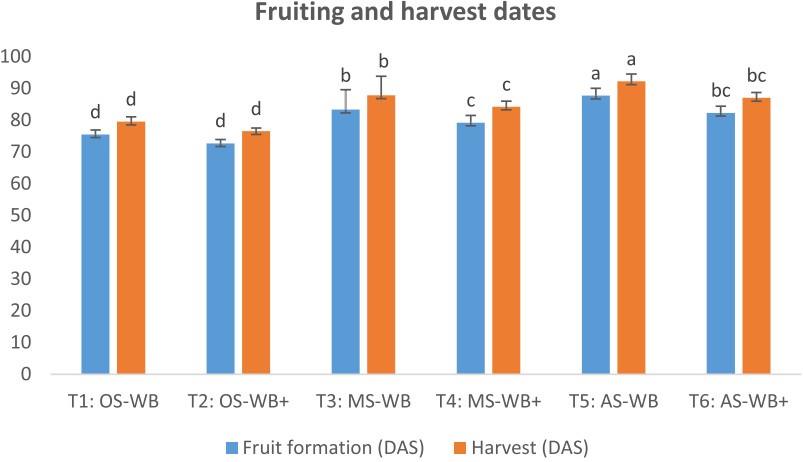

**Figure 3 Dates of fruit formation and harvest of experimental treatments.** OS, oak sawdust; MS, maple sawdust; AS, apple sawdust; different letters indicate a statistically significant difference (at *P* value < 0.05) for each indicator according to Duncan's multiple range test.

## Shiitake production in different treatments

Results in Table 2 showed that the separate effect of the factor 'substrate' and the combined effect of both factors 'substrate x supplementation' were statistically significant ($P_{value}$ < 0.05) on MN1 and MN2, MW1 and MW2, and the biological yield obtained at the second harvest (BY2). However, the separate effect of the factor 'supplementation' was significant on all productive indicators except MN2.

Compared to T1, the average mushroom number was significantly lower, and the average mushroom weight was significantly higher in all treatments at both harvests.

At the first harvest, MN1 decreased significantly by 18.2 and 4.9 mushrooms on average, and MW1 increased by 30.4 and 15.4 grams in supplemented oak and apple sawdust (T2 and T6), respectively, compared to non-supplemented substrates (T1 and T5). At the same harvest, average BY1 had a significant improvement in all supplemented oak, maple, and apple sawdust compared to non-supplemented ones (by around 21.5, 11.4, and 20.3% in T2, T4, and T6, respectively).

At the second harvest, MN2 increased significantly in supplemented maple and apple sawdust compared to non-supplemented substrates (by 3.7 mushrooms on average in T4 and by 8.0 mushrooms in T6). At the same harvest, MW2 increased by around 16.5 and 2.9 g in supplemented oak and maple sawdust (T2 and T4) compared to non-supplemented substrates (T1 and T3). At this harvest, BY2 showed a significant increase in supplemented maple and apple sawdust compared to non-supplemented cases (by around 36.4 and 27.4% in T4 and T6, respectively).

The highest BE was recorded on the supplemented oak sawdust substrate (T2: 59.7%). Supplementation caused a significant improvement in BE in all supplemented *vs.* non-supplemented sawdust. The BE obtained in supplemented maple sawdust (T4) was comparable to T1 (53.5%), although that obtained in non-supplemented maple sawdust (T3) was significantly lower than that of T1 (42.2%).

**Table 3 Variation in productive indicators between harvests 1 and 2.**

|         | T1: OS-WB       | T2: OS-WB+      | T3: MS-WB       | T4: MS-WB+       | T5: AS-WB       | T6: AS-WB+      |
|---------|-----------------|-----------------|-----------------|------------------|-----------------|-----------------|
| **MN1** | 29.5 ± 3.5a     | 11.3 ± 2.2a     | 13.0 ± 2.1a     | 15.3 ± 2.0a      | 10.3 ± 2.7ab    | 6.3 ± 2.7a      |
| **MN2** | 24.0 ± 2.8b     | 9.3 ± 2.3a      | 9.5 ± 1.9b      | 13.2 ± 2.8a      | 4.8 ± 1.7       | 12.8 ± 4.8b     |
| *P value* | 0.014*        | 0.155           | 0.012*          | 0.150            | 0.002*          | 0.021*          |
| **MW1** | 8.8 ± 0.5a      | 39.2 ± 1.7a     | 21.7 ± 3.0a     | 20.6 ± 2.1a      | 20.9 ± 1.6a     | 36.34 ± 2.6a    |
| **MW2** | 11.6 ± 3.3a     | 28.1 ± 1.2b     | 14.6 ± 0.5b     | 17.5 ± 2.7a      | 22.4 ± 2.6a     | 20.3 ± 1.3b     |
| *P value* | 0.091         | 0.000*          | 0.002*          | 0.05             | 0.246           | 0.000*          |
| **BY1** | 275.0 ± 52.2a   | 350.1 ± 28.9a   | 291.8 ± 35.1a   | 329.7 ± 10.3a    | 249.5 ± 55.9a   | 312.8 ± 27.6a   |
| **BY2** | 322.0 ± 43.7a   | 322.1 ± 39.8a   | 130.6 ± 32.5b   | 205.6 ± 30.68b   | 103.6 ± 2.1b    | 142.8 ± 5.3b    |
| *P value* | 0.122         | 0.185           | 0.000*          | 0.000*           | 0.001*          | 0.000*          |

Note:
Means followed by different letters in the same column indicate a significant difference at *P* value < 0.05 according to independent sample t-test. OS, oak sawdust; MS, Maple sawdust; AS, Apple sawdust; WB, Wheat bran; +, nano-amino supplement added to the substrate; MN1, mushroom number in harvest 1; MN2, mushroom number in harvest 2; MW1, mushroom weight in harvest 1 (g); MW2, mushroom weight in harvest 2 (g); BY1, biological yield (g/kg) at harvest 1; BY2, biological yield (g/kg) at harvest 2. Different letters in the same column indicate a statistically significant difference according to Duncan's multiple range test at *P* value < 0.05. An asterisk (*) indicates a statistically significant effect of the studied factors on tested indicators.

## Variation in production in each treatment

Comparing the variation in productive indicators between the first and second harvests (Table 3), it was found that the average MN2 significantly decreased compared to the average MN1 in T1 (by 5.5), T3 (by 3.5), and T5 (by 5.5), while it increased significantly in T6 (by 6.5). MW2 decreased significantly with respect to the treatments T2 (by 11.1 g), T3 (by 7.1 g), and T6 (by 16.0 g). Also, the biological yield showed a significant decline in the second harvest in T3, T4, T5, and T6 (by 55.2, 37.6, 58.4, and 54.3%, respectively). Therefore, production was more consistent among consecutive harvests only on substrates containing oak sawdust, while it decreased progressively throughout consecutive harvests on maple and apple sawdust, whether supplemented or not.

## Mushrooms' physical characteristics

Results in Table 4 showed that the factor 'substrate' had a significant effect (*P* value < 0.05) on pileus diameter and mushroom firmness, while the factor 'supplementation' had only a significant effect on pileus diameter. The combined factors significantly affected pileus thickness and the ratio PD/SL. PD was significantly higher in T2 compared to all treatments and increased by 1.4 cm compared to T1. PT increased significantly in T4 compared to T3 and T1 (1.6, 1.0, and 1.2 cm, respectively). The SD was comparable on all tested substrates and ranged from 0.6 to 1.2 cm. The ratio PD/SL was significantly lower in T6 compared to control (1.5 and 1.9, respectively). Mushroom firmness was significantly lower in T5 compared to T2 and T4, but it was comparable to control in all treatments, ranging from 4.2 to 5.5 mm.

## Correlation of total biological yield and productive indicators

Results of stepwise regression (Table 5) showed that the TBY in T1, T5, and T6 was the most positively affected by the variation in the biological yield obtained at the first harvest

**Table 4 Shiitake physical characteristics obtained at different treatments.**

| Treatments | PD (cm) | PT (cm) | SD (cm) | SL (cm) | PD/SL | Firmness (mm) |
|---|---|---|---|---|---|---|
| T1: OS-WB | 5.3 ± 1.3b | 1.2 ± 0.0bc | 0.9 ± 0.2a | 3.2 ± 0.8ab | 1.9 ± 0.4a | 5.2 ± 0.8abc |
| T2: OS-WB+ | 6.7 ± 0.7a | 1.3 ± 0.1abc | 1.1 ± 0.3a | 3.6 ± 0.5a | 1.8 ± 0.2ab | 5.5 ± 0.7a |
| T3: MS-WB | 4.4 ± 0.6b | 1.0 ± 0.5c | 0.6 ± 0.2a | 2.7 ± 0.6b | 1.6 ± 0.2ab | 5.1 ± 0.9abc |
| T4: MS-WB+ | 5.0 ± 0.8b | 1.6 ± 0.4a | 1.2 ± 0.9a | 2.8 ± 0.8ab | 1.8 ± 0.1ab | 5.3 ± 0.9ab |
| T5: AS-WB | 5.0 ± 1.0b | 1.5 ± 0.3ab | 1.2 ± 0.7a | 3.0 ± 0.8ab | 2.0 ± 0.4a | 4.2 ± 0.9c |
| T6: AS-WB+ | 5.3 ± 1.0b | 1.4 ± 0.2abc | 1.2 ± 0.5a | 3.4 ± 0.4ab | 1.5 ± 0.2b | 4.4 ± 0.7bc |
| *P value* | | | | | | |
| Sub | 0.007* | 0.300 | 0.330 | 0.081 | 0.495 | 0.007* |
| Sup | 0.018* | 0.070 | 0.133 | 0.163 | 0.152 | 0.331 |
| Sub x sup | 0.288 | 0.043* | 0.367 | 0.722 | 0.030* | 0.967 |

Note:
OS, oak sawdust; MS, Maple sawdust; AS, Apple sawdust; WB, Wheat bran; +, nano-amino supplement added to the substrate; PD, pileus diameter; PT, pileus thickness; SD, stipe diameter; SL, stipe length. Different letters in the same column indicate a statistically significant difference according to Duncan's multiple range test at $P$ value <0.05. An asterisk (*) indicates a statistically significant effect of the studied factors on tested indicators

**Table 5 Significant correlations between the total biological yield (dependent variable) and productive indicators (independent variables) in the tested treatments ($n = 6$).**

| | Dep | Ind | Equation | Adj R2 |
|---|---|---|---|---|
| T1: OS-WB | TBY | BY1 | 117.349 + (0.977 × BY1) | 0.94 |
| T2: OS-WB+ | TBY | PD | 212.174 + (0.924 × PD) | 0.82 |
| T4: MS-WB+ | TBY | MW2 | 383.627 + (0.982 × MW2) | 0.96 |
| T5: AS-WB | TBY | BY1 | 103.669 + (0.999 × BY1) | 0.99 |
| T6: AS-WB+ | TBY | BY1 | 139.152 + (0.982 × BY1) | 0.96 |

Note:
Dep, dependent variable; Ind, independent variable(s); TBY, total biological yield; BY1, biological yield of harvest 1; BY2, biological yield of harvest 2; FW1, fruit weight of harvest 1; FW2, fruit weight of harvest 2; OS, Oak sawdust; MS, maple sawdust; AS, apple sawdust; WB, wheat bran; +, nano-amino supplement added to the substrate.

($R^2$ = 0.94, 0.99, and 0.96, respectively). In T2, TBY was positively affected by the PD ($R^2$ = 0.82); thus, a higher PD obtained at the first harvest following the application of nano-amino will eventually increase the total production on oak sawdust substrate. In T4, TBY was positively affected by the MW2 ($R^2$ = 0.96), meaning that heavier mushrooms obtained at the second harvest will enhance the total production of supplemented maple sawdust.

## Shiitake nutritional value

Analysis of the mushrooms' nutritional composition (Table 6) showed that on non-supplemented substrates, shiitake protein content was the highest on AS, followed by MS and OS (9.2, 8.2, and 2.7% in T1, T3, and T5, respectively). Supplementation significantly increased protein content in mushrooms cultivated on the three supplemented sawdust types (5.6, 9.5, and 10.9% in T2, T4, and T6) compared to non-supplemented ones (2.7, 8.2, and 9.2% in T1, T3, and T5). Mushrooms' crude fiber content increased significantly on supplemented OS and MS compared to

**Table 6 Composition of Shiitake mushrooms cultivated on supplemented *vs.* non-supplemented substrates.**

| Treatments | Protein (%) | Fibers (%) | Fat (%) | Ash (%) | Carbs (%) | Vit C (mg/100 g) | Vit D (µg/L) |
|---|---|---|---|---|---|---|---|
| **T1: OS-WB** | 2.7 ± 0.15e | 1.5 ± 0.1d | 0.3 ± 0.1a | 0.6 ± 0.1c | 7.3 ± 0.1a | 4.5 ± 0.1c | <3 |
| **T2: OS-WB+** | 5.6 ± 0.3d | 3.6 ± 0.3a | 0.4 ± 0.07a | 0.9 ± 0.2b | 4.2 ± 1.0c | 4.8 ± 0.4c | <3 |
| **T3: MS-WB** | 8.2 ± 0.3c | 3.1 ± 0.2b | 0.4 ± 0.02a | 1.0 ± 0.1b | 2.8 ± 0.5de | 8.0 ± 0.1b | <3 |
| **T4: MS-WB+** | 9.5 ± 0.3b | 3.8 ± 0.2a | 0.3 ± 0.07a | 1.1 ± 0.1b | 1.9 ± 0.7e | 8.3 ± 0.1b | <3 |
| **T5: AS-WB** | 9.2 ± 0.1b | 3.7 ± 0.3a | 0.3 ± 0.08a | 1.1 ± 0.2b | 5.4 ± 0.5b | 11.0 ± 0.4a | <3 |
| **T6: AS-WB+** | 10.9 ± 0.3a | 2.5 ± 0.2c | 0.4 ± 0.07a | 1.4 ± 0.1a | 3.2 ± 0.6cd | 11.2 ± 0.2a | <3 |
| *P value* | 0.000* | 0.000* | 0.293 | 0.000* | 0.000* | 0.000* | |

**Note:**
OS, oak sawdust; MS, Maple sawdust; AS, Apple sawdust; WB, Wheat bran; +, nano-amino supplement added to the substrate; carbs, carbohydrates. Different letters in the same column indicate a statistically significant difference according to Duncan's multiple range test at *P* value < 0.05. An asterisk (*) indicates a statistically significant effect of the studied factors on tested indicators.

non-supplemented cases. On the contrary, it decreased in supplemented than in non-supplemented AS treatments. The fiber content of mushrooms ranged between 1.5 and 3.8%. Fiber content was ameliorated on MS and AS (supplemented or not) compared to the commercially applied oak sawdust-based substrate. Crude fat did not differ significantly among treatments and ranged between 0.3 and 0.4%. Further, ash contents obtained on supplemented OS and AS were significantly higher than those recorded on non-supplemented substrates (0.9 and 1.4% in T2 and T6 compared to 0.6 and 1.1% in T1 and T5, respectively). On the other hand, there was a slight or significant decrease in mushrooms' carbohydrate contents obtained from supplemented *vs.* non-supplemented substrates. Though vitamin C contents did not differ because of substrate supplementation, they were significantly higher in mushrooms cultivated on substrates containing AS (11.0–11.2 mg/100 g) than those containing MS (8.0–8.3 mg/100 g) or OS (4.5–4.8 mg/100 g). Vitamin D content showed similar values (<3 µg/L) in all treatments.

# DISCUSSION

In the current study, the growth of shiitake 3782 took around 72 DAS to complete the mycelia run on oak sawdust. Our results were comparable to *Chen (2005a)* who mentioned that the complete mycelium run on oak sawdust ranged from about 30 to 120 DAS, depending on the shiitake strain under investigation. Also, oak sawdust showed earlier growth than other sawdust types, presumably because of its lower pH and higher ash content. Shiitake mushrooms prefer a slightly acidic pH range from about 5.5 to 6.5 during spawn run (*Bruhn, Mihail & Pickens, 2009*; *Zied, Savoie & Pardo-Giménez, 2011*), and *Chen (2005a)* reported that an abundance of minerals in the substrate promotes mycelial growth and fruit formation. Further, the C/N ratio of the substrate has a major influence on mycelium growth and fruit body formation of shiitake, thus on yields (*Hoa, Wang & Wang, 2015*; *Bellettini et al., 2019*). The minimum C/N ratio required for shiitake growth is 25:1, the highest is 55: 1 (*Zied, Savoie & Pardo-Giménez, 2011*), and the optimum C/N ratio is 40:1 (*Chen, 2005b*). Also, according to *Desisa et al. (2023)*, a lower C/N ratio is favored during the mycelia development of shiitake. However, current findings confirm

that shiitake grew at higher C/N ratios than those reported earlier, but complete mycelial development was achieved first on oak sawdust with a lower C/N ratio compared to maple and apple sawdust.

In term of substrate degradation, fungi degrade the insoluble lignocellulosic components of substrates *via* enzymatic pathways (*Abou Fayssal et al., 2021a*), and substrate properties play a major influence on the substrate decomposition process, thus on mycelial growth throughout the substrate (*Picornell-Buendía, Pardo & De Juan, 2015*; *El Sebaaly, Najjar & Sassine, 2021*). Lignin, hemicellulose, and cellulose serve as energy sources for fungal growth because they contain carbon, hydrogen, and oxygen. They are degraded progressively along the cultivation cycle (*Andrade et al., 2010*). Lignin is the most recalcitrant component of the plant cell wall, and its effect on the bioavailability of other cell wall components is primarily a physical restriction. Therefore, the higher the proportion of lignin in the initial substrate mixture, the lower the substrate's bioavailability (*Haug & Haug, 1993*). During the vegetative phase of mycelia growth, lignin is degraded first to allow access to holocellulose (hemicellulose and cellulose) (*Kurt & Büyükalaca, 2010*; *Xiao et al., 2017*). Further, according to *Ratnanindha, Sutapa & Irawati, 2019*, mycelia growth is negatively correlated with the hemicellulose content of the growing medium. Although oak sawdust contained the highest amounts of lignin, it was the poorest in hemicellulose and had a lower content of holocellulose compared to maple and apple sawdust, which could explain the faster growth of shiitake on the former substrate.

Furthermore, supplementation of the growing substrates using 5 g/L of Lithovit-®Amino 25 induced earliness in harvest dates ranging between 3.0 and 5.2 d in supplemented substrates compared to non-supplemented cases. Earlier, the same product applied at spawning caused an early harvest of 2.3 and 3.3 d when used with 3 and 5 g/L, respectively, on *Pleurotus ostreatus* (*Naim et al., 2020b*). Nitrogen plays a regulatory role during the enzymatic breakdown of the substrate lignocellulose (*Nagy et al., 2015*) and is relatively low in wood (*Moore et al., 2021*). Nanoparticles possess a large surface area and can hold plenty of nutrients, which they release gradually to the growing substrate (*Gade et al., 2023*). Lithovit-®Amino 25 is rich in amino acids, which were reported to play a major role in transporting nitrogen into the living cells of fungus (*Mikeš et al., 1994*). Amino acids are essential for mycelial maturation (*Du et al., 2019*), thus for the proper initiation of the fruiting stage. Also, the product is rich in micronutrients that were reported to affect the fruit body induction of several fungal species (*Sales-Campos et al., 2009*). Therefore, the different components contained in this supplement could have positively induced earlier fruit formation in the supplemented substrates. As a matter of fact, besides environmental factors such as cooling, light, and humidity, enzymatic activity was found to be correlated with fruit induction in various fungal species (*Terashita et al., 1998*; *Deacon, 2006*; *Sakamoto, 2018*). However, future investigations are needed to clearly point out the effect of supplementation at soaking on the enzymatic activities intervening with shiitake fruit body development.

In this study, non-supplemented oak and maple sawdust had lower averages of number and weight of fruiting bodies than the values reported by *Ranjabar & Olfati (2016)* on
similar substrates. Supplemented oak and apple sawdust with 5 g/L Lithovit®-Amino 25 decreased mushroom number and increased mushroom weight at the first harvest, while the same product applied to oyster mushroom (3 g/L) increased fruit number and weight (*Naim et al., 2020b*). On an industrial scale, farmers can harvest 0.3 to 0.5 kg of fresh shiitake mushrooms from 1 kg of dried substrate (30–50% BE) (*Chen, 2005a*). Therefore, biological yields obtained from oak (supplemented or not), supplemented maple sawdust, and supplemented apple sawdust (around 0.6, 0.4, and 0.3 kg of mushrooms per 1 kg of substrate) were good enough to adopt the three types of hardwood sawdust obtained from local agro-forests for shiitake production. On an experimental scale, higher BE values were reported which can reach 90% (*Fan & Soccol, 2005*). Values of BE obtained on non-supplemented oak and apple sawdust (59.7 and 35.3%, respectively) were lower than those reported in *Yu et al. (2022)*, which were of 80.79% and 81.7%, respectively. On maple logs, *Ranjabar & Olfati (2016)* obtained a BE of 79.5%. Also, in the current study, higher biological yields were obtained from oak than maple and apple sawdust probably because of the higher C/N ratios of the latter substrates. A high C/N (more C) can limit the availability of N, leading to slow growth and low mushroom yields (*Desisa et al., 2023*). Improvement in total biological yield because of supplementation was of 9.8, 21.0, and 22.5%, respectively on oak, maple, and apple sawdust. Therefore, supplementation was more effective on maple and apple sawdust than on oak sawdust, although only supplemented maple sawdust reached a comparable production to that obtained on the commercially used oak sawdust-based substrate. According to *Zied, Savoie & Pardo-Giménez (2011)* supplementation using nitrogen-rich additives reduces the levels of carbon in the substrate mainly because of the use of bulk materials. For instance, nitrogen added in an organic form, such as soybean meal, cotton seed meal, peanut meal (*Kapoor, Pardeep & Katyal, 2009*) chicken manure (*Desisa et al., 2023*), or mineral form (*Queiroz, Marino & Eira, 2004*) caused significant improvement in shiitake yield. Lithovit®-Amino25 contained both types of nitrogen, organic and mineral, which could have replenished the initially low nitrogen content in maple and apple sawdust, causing amelioration in shiitake yields obtained on both substrates.

In term of mushroom's physical characteristics, earlier, *Baktemur et al. (2022)* found the following physical characteristics for shiitake mushrooms cultivated on oak sawdust: cap diameter 5.1 cm, cap thickness 1.3 cm, and stipe length 2.7 cm. Also, on oak sawdust, *Moonmoon et al. (2011)* obtained shiitake mushrooms with a pileus diameter of 6.0 cm, pileus thickness 1.0 cm, stipe length 4.5 cm, and stipe diameter 1.0 cm. These values were close to those obtained in the current study. The increase (whether significant or slight) in pileus diameter and mushroom firmness obtained on supplemented substrates is an amelioration of quality. Eventually, the pileus gives the mushroom a more substantial bite, especially if it is intended to be eaten whole, and mushrooms having larger pileus are more acceptable at the market (*Synytsya et al., 2008*).

Mushroom composition is largely affected by substrate composition (*El Sebaaly et al., 2019*), and by the type of supplements added to the growing substrate (*Abou Fayssal et al., 2021b*). Protein content of shiitake mushrooms obtained in the present study ranged between 2.7% and 10.9%, lower than the ranges obtained in earlier studies 12.4–22.6% on

various substrates (*Gaitán-Hernández et al., 2005*; *Moonmoon et al., 2011*; *Baktemur et al., 2022*). Protein is the most critical component contributing to the nutritional value of the mushroom (*El Sebaaly, Najjar & Sassine, 2021*). Similarly, to results obtained on *Pleurotus ostreatus* (*Naim et al., 2020b*), supplementation increased protein content of shiitake mushrooms. Nowadays, countless efforts are made to search for new and sustainable protein sources to fill the nutritional needs of a growing world population (*Avelar et al., 2022*). Cultivation of shiitake on oak, maple, and apple sawdust allows the bioconversion of agro-forest wastes into protein-rich food, and supplementation using Lithovit®-Amino 25 adds the advantage of improving protein content in the produced mushrooms. The fiber content of mushrooms in the current study was lower than the range indicated by *Desisa et al. (2023)*. Twenty-five percent of the daily required intake of dietary fiber can be obtained from eating edible mushrooms (*Cheung, 2013*). Most of the mushroom dietary fibers are water-insoluble, such as chitin and β-glucans (*Zhao et al., 2022*). Both components have many applications in biomedicine because of their anti-inflammatory, anti-obesity, anti-allergic, anti-osteoporotic, anticancer, and immunomodulatory effects (*Ali Komi, Sharma & Dela Cruz, 2018*; *Bhoite, Satyavrat & Premasudha Sadananda, 2022*).

## CONCLUSIONS

The work is the first to investigate the effect of nano-supplementation on shiitake mushrooms cultivated on locally available lignocellulosic substrates, oak, maple, and apple sawdust. Oak sawdust showed more consistent production over two consecutive harvests than apple and maple sawdust. Supplementation at first soaking ameliorated the production at the second harvest on maple and apple sawdust. Therefore, under these experimental conditions, only supplemented maple substrate can be recommended for use as an alternative to oak sawdust for the commercial production of shiitake. However, since the product ameliorates yields while improving the mushroom's nutritional value through higher protein and crude fiber contents, it is recommended to test its effects at other timing of application, such as at spawning to overcome the delay in mycelia run and fruiting on maple compared to oak sawdust, or at the second soaking (after the first harvest) to increase the number of harvests per substrate, block substrate produced comparable yields to those obtained on the commercially used oak sawdust substrate.

### Funding
This work was supported by the BULGARIAN DEVELOPMENT AID (Grant no. 1), Bulgarian Embassy in Lebanon. The funders had no role in study design, data collection and analysis, decision to publish, or preparation of the manuscript.

### Grant Disclosures
The following grant information was disclosed by the authors:
BULGARIAN DEVELOPMENT AID (Grant no. 1).
Bulgarian Embassy in Lebanon.

## Competing Interests

The authors declare that they have no competing interests.

## Author Contributions

- Stephanie Nabhan performed the experiments, analyzed the data, prepared figures and/or tables, authored or reviewed drafts of the article, and approved the final draft.
- Soukayna Haidar Ahmad performed the experiments, analyzed the data, prepared figures and/or tables, and approved the final draft.
- Zeina El Sebaaly conceived and designed the experiments, prepared figures and/or tables, authored or reviewed drafts of the article, and approved the final draft.
- Teodor Nedelin conceived and designed the experiments, authored or reviewed drafts of the article, and approved the final draft.
- Youssef Najib Sassine conceived and designed the experiments, authored or reviewed drafts of the article, and approved the final draft.

## Data Availability

The raw data is available in the Supplemental File.

## Supplemental Information

Supplemental information for this article can be found online at http://dx.doi.org/10.7717/peerj.18622#supplemental-information.

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
