# Peer review of "A first trial to supplement local hardwood sawdust at the first soaking for the cultivation of Shiitake (Lentinula edodes)"

_PeerJ, doi:10.7717/peerj.18622_

## Round 0.1 · original submission · Minor Revisions

· Academic Editor

Minor Revisions

Please, authors kindly attend to the comments raised by the reviewers.

Make effort to provide strong detail, not only in the manuscript, but also in the responses to their respective comments. I look forward to your revised manuscript. Thank you

Reviewer 1 ·

Basic reporting

The writing of the article is sloppy. However, when the study was examined, good results were obtained. I think it will make an important contribution to the literature.

Experimental design

Research question well defined, relevant & meaningful. It is stated how research fills an identified knowledge gap.

Rigorous investigation performed to a high technical & ethical standard.

Methods described with sufficient detail & information to replicate.

Validity of the findings

All underlying data have been provided; they are robust, statistically sound, & controlled.

Conclusions are well stated, linked to original research question & limited to supporting results.

Additional comments

The writing of the article is sloppy. However, when the study was examined, good results were obtained. I think it will make an important contribution to the literature.

Annotated reviews are not available for download in order to protect the identity of reviewers who chose to remain anonymous.

Reviewer 2 ·

Basic reporting

no comment

Experimental design

no comment

Validity of the findings

no comment

Additional comments

This manuscript studied the effect of different substrate on the production and nutrient profile of shiitake mushroom. The correlation of total biological yield and productive indicator was revealed. This study provided evidence for uplifting the quality of shiitake mushroom using substrate supplementary. Some questions in this manuscript needs to be reconsidered. This manuscript is suggested to be published after minor revision.
1. Abstract: the description of the substrate formula was suggested to be cleared in materials and methods. The improvement of mushroom nutrition should also be listed, and the improvement of production was suggested to be briefer.
2. The format of paragraph in the whole manuscript should be reset.
3. Line 54: the year should be checked, line 56: the format of word should be revised, line 62: comma should be added before “but”.
4. Materials and methods: Line 210: the calculation equation should be checked. The subscript letter in the equations should be marked clearly.
5. Results: The abbreviation can be used directly if it had been cleared in the materials and methods. Line 319: the data here should be checked.
6. The format of the tables should be revised. The significance analysis annotation in the table 4 should be completed. The superscript and subscript in table 7 should be unified.
7. The citation format in the whole manuscript should be revised, “et al” should be supplemented.
8. The units in the whole manuscript should be revised according to SI standard.
9. The spelling should be checked. Line 420: “pi-leus” should be revised. Line 443: “wa-ter” should be revised, and the abbreviation IDF and SDF were not in line with the former words.
10. Line 440-446, the description about the benefit of fiber was not suitable here, the reason of mushroom nutrient improvement should be supplemented.
11. Conclusions: The significance and prospect of this study should be introduced.

Reviewer 3 ·

Basic reporting

The manuscript is interesting. Some pictures of mushroom cultivation and fruiting body should be presented.

Experimental design

The manuscript is in scope of Biological Science. The experiment is well planned. How did the author measure the firmness of mushroom? Please clearly describe.

Validity of the findings

The finding is new for Shiitake but not new for mushroom cultivation. However this finding should be useful for Shiitake cultivation in the future. It should be clearly discussed on the economic effect of use this supplement in mushroom cultivation.

Additional comments

1. The title should be revised because effect of substrate was also investigated.
2. Picture of mushroom cultivation and fruiting body should be included.

Reviewer 4 ·

Basic reporting

1. The research is interesting and enlightening however, there are long sentences that could be re-phrased e.g lines 82-86, 92-95 in Introduction.

2. Days could be rounded off to the nearest whole number for instance line 28.

3. Line 39, apostrophe not required.

4. Keyword suggestion: Delete additives, amino acid, add nano-supplement

5. Line 80, Do you mean mushroom colonization at the end of spawn run or mushroom fructification?

Experimental design

Yes

Validity of the findings

Yes

Annotated reviews are not available for download in order to protect the identity of reviewers who chose to remain anonymous.

---

## Round 0.2 · Minor Revisions

· Academic Editor

Minor Revisions

Thank you authors for your patience. You can see reviewers have very positive feedback for acceptance. Please, kindly attend to the very minor corrections identified by one reviewer.

Reviewer 1 ·

Basic reporting

There are no problems after corrections

Experimental design

There are no problems after corrections

Validity of the findings

There are no problems after corrections

Additional comments

There are no problems after corrections

Reviewer 2 ·

Basic reporting

I am glad that the majority of questions we proposed had been revised. I recommend this article to be published. However, a few questions should be considered. The subscript letter in the quations is able to be added in Word, for example quations in following articles published in peerj (https://doi.org/10.7717/peerj.17943
https://doi.org/10.7717/peerj.18238
https://doi.org/10.7717/peerj.18261
https://doi.org/10.7717/peerj.17984). As subscript letter was used in the text.

Experimental design

no comment

Validity of the findings

no comment

Reviewer 3 ·

Basic reporting

No comment

Experimental design

No comment

Validity of the findings

no comment

Additional comments

no comment

Reviewer 4 ·

Basic reporting

.

Experimental design

.

Validity of the findings

.

Additional comments

1. Lines 65, 66: Close the gap
2. Line 274: Replace ‘recorded’ with ‘had’. Note the parameter cannot record itself.
3. Line 338: Add ‘’The growth of’’ Shiitake ...
4. Line 347: 55:??? Complete
5. Table 5: For consistency sake, reconcile harvest and flush throughout the manuscript for instance Fig. I, lines 231, 271, 277, 281.
6. Fig. I:
• Delete grids
• Change the fig. Title to ‘’ Dates of fruits formation and harvest on experimental treatments’’
• Line 263: Factor was mentioned here, was the research a factorial experiment and was it analysed as such? This is the first time –factor- was mentioned.
7. Align text using Justify

---

## Round 0.3 · accepted · Accept

· Academic Editor

Accept

Dear Authors,

Thank you for revising your work, and considering all comments raised. It is now acceptable for publication. We appreciate your finding PeerJ as your journal of choice, and looking forward to your future scholarly contributions.

Congratulations